# Genes, Structural, and Biochemical Characterization of Four Chlorophyllases from *Solanum lycopersicum*

**DOI:** 10.3390/ijms231911716

**Published:** 2022-10-03

**Authors:** Guangyuan Liu, Xue Meng, Yujun Ren, Min Zhang, Ziqing Chen, Zhaoqi Zhang, Xuequn Pang, Xuelian Zhang

**Affiliations:** 1College of Life Sciences, South China Agricultural University, Guangzhou 510642, China; 2Fujian Provincial Key Laboratory of Plant Functional Biology, College of Life Sciences, Fujian Agriculture and Forestry University, Fuzhou 350002, China; 3College of Horticulture, South China Agricultural University, Guangzhou 510642, China

**Keywords:** chlorophyllase, biochemical characteristics, catalytic triad, substrate specificity

## Abstract

Recent studies have confirmed that chlorophyllase (CLH), a long-found chlorophyll (Chl) dephytylation enzyme for initiating Chl catabolism, has no function in leaf senescence-related Chl breakdown. Yet, CLH is considered to be involved in fruit degreening and responds to external and hormonal stimuli. The purpose of this work was to elucidate in detail the biochemical, structural properties, and gene expression of four CLHs from the *Solanum lycopersicum* genome so as to understand the roles of *Solanum lycopersicum* chlorophyllases (SlCLHs). *SlCLH1/4* were the predominantly expressed *CLH* genes during leaf and fruit development/ripening stages, and *SlCLH1* in mature green fruit was modulated by light. SlCLH1/2/3/4 contained a highly conserved GHSXG lipase motif and a Ser-Asp-His catalytic triad. We identified Ser159, Asp226, and His258 as the essential catalytic triad by site-directed mutagenesis in recombinant SlCLH1. Kinetic analysis of the recombinant enzymes revealed that SlCLH1 had high hydrolysis activities against Chl a, Chl b, and pheophytin a (Phein a), but preferred Chl a and Chl b over Phein a; SlCLH2/3 only showed very low activity to Chl a and Chl b, while SlCLH4 showed no Chl dephytylation activity. The recombinant SlCLH1/2/3 had different pH stability and temperature optimum. Removal of the predicted N-terminal processing peptide caused a partial loss of activity in recombinant SlCLH1/2 but did not compromise SlCLH3 activity. These different characteristics among SlCLHs imply that they may have different physiological functions in tomato.

## 1. Introduction

Chlorophylls (Chls), which consist of a porphyrin ring containing a central Mg cation and a phytol chain, are the essential components for capturing light energy to drive photosynthesis [1]. Given the pivotal role of the phytol chain in anchoring Chls into thylakoid membranes and the loss of the phytol chain is considered a prerequisite for further Chl degradation by downstream enzymes [2], the process of Chl dephytylation during chlorophyll breakdown has received much attention for decades. To date, the authentic enzymes responsible for Chl dephytylation are chlorophyll dephytylase 1 (CLD1), pheophytinase (PPH), and chlorophyllase (CLH) [3]. PPH expression peaks during leaf senescence, and it is considered to be the major dephytylation enzyme active during leaf senescence [2]. By contrast, CLD1 is expressed mainly in green tissue and functions in Chl turnover at a steady state [4]. Compared to the dephytylation roles assigned to PPH and CLD1, the role of CLHs is still a puzzle.

Since it was first reported more than a century ago, the physiological and biochemical characteristics of CLHs from various plants and algal species, including *Chenopodium album* [5], *Citrus limon* [6,7], *Pachira macrocarpa* [8], *Chlamydomonas reinhardtii* [9], *Oscillatoria acuminata* PCC6304 [10], and *Arabidopsis thaliana* [11], have been intensively studied. CLH was considered to localize in chloroplasts and the thylakoid membrane because endogenous Chl hydrolyzation only takes place when chloroplast membranes are disrupted [12,13,14]. In fact, CLHs from *Ginkgo biloba* [15] and *Citrus limon* [6] were demonstrated to localize in the chloroplast. Surprisingly, not all chloroplasts contain predicted CLHs, such as CaCLH (*Chenopodium album* CLH) and AtCLH1/2 (Arabidopsis CLH1/2) [16,17], so whether CLHs are truly involved in the decomposition of Chl remains questionable.

CLHs have been implicated in the postharvest senescence of broccoli (*Brassica oleracea* var *italica*) and ethylene-induced citrus (*Citrus* spp.) fruit ripening [6,18,19], whereas the long-believed role of CLHs in hydrolyzing Chl in senescing leaves has been challenged by genetic studies in Arabidopsis because the phenotype of loss-of-function double mutants of Arabidopsis *CLH1* and *CLH2* genes showed no defects in senescence-related Chl breakdown compared with wild type [16]. Instead, PPH has been identified as an enzyme that catalyzes senescence-related Chl dephytylation in vivo [2]. Interestingly, leaves in tomato *SlPPH*-silenced transgenic lines show a stay-green phenotype, whereas chlorophyll in fruit is eventually degraded [20], suggesting that other hydrolases are involved in fruit ripening. It is worth noting that the expression profile of *SlCLH1* is correlated with Chl content during fruit ripening [21], implying that SlCLHs may participate in dephytylation during fruit ripening.

According to phenetic analysis, *CLH* genes are usually divided into two groups in the analyzed species. The existence of two groups of CLHs suggests functional diversification and distinct selective pressures [21]. Researchers have indicated that Group II CLHs are involved in the homeostasis of Chls in leaves [22], while Group I CLHs are responsive to external and hormonal stimuli, such as wounding [23,24], pathogen elicitors [24], abscisic acid (ABA) [25], etc. Most recently, a report revealed additional information on CLHs. It was found that Arabidopsis AtCLH1, which belongs to group I CLH, is located in developing chloroplasts but not in mature chloroplasts and played a role in protecting young leaves from long-term photodamage by facilitating FtsH-mediated D1 degradation [11]. Although there is a long research history on the underlying characteristics and function of CLH, much is still unknown about CLH [26].

Tomato is a typical model plant. Besides a rather short life cycle, it offers established genetic tools to allow the simultaneous analysis of dephytylation during leaf senescence, fruit ripening, and other physiological processes. Though four putative *S. lycopersicum* CLH sequences were annotated in the genome, molecular structure, substrate specificity, enzyme kinetics, and biochemical characteristics of SlCLHs have not yet been investigated. In this study, we analyzed the expression and biochemical characteristics of these four CLHs. This study aims to provide a detailed biochemical characterization of the properties of SlCLHs to better understand the potential molecular and physiological basis of CLHs in Chl degradation during fruit ripening or other developmental stages.

## 2. Results

### 2.1. Differential Expression Pattern of Four SlCLH Genes in Various Tissues

The transcriptome data of *SlCLHs*, *S. lycopersicum* cv. Heinz and *S. lycopersicum* M82, were extracted from the International Tomato Genome Sequencing Consortium (SGN; solgenomics.net) database (version ITAG 2.4) during plant development and fruit ripening, respectively (Figure 1A), and the relative transcript abundances were subsequently validated by qRT-PCR (Figure 1B,C). Compared to *SlCLH2/3/4*, *SlCLH1* displays a relatively high expression during leaf development and fruit development/ripening (Figure 1A). In leaves, *SlCLH1* expression was higher at the young and mature stage, and significantly decreased in aging leaves thereafter (*p* < 0.001) (Figure 1B). *SlCLH1* expression during fruit ripening exhibits similar behavior to that observed during leaf senescence (Figure 1C). *SlCLH1* exhibited the highest level of mRNA accumulation at 30 DPA and the mature green fruit stage and then declined during ripening. The correlation between *SlCLH1* expression pattern and Chl content indicated its involvement in Chl homeostasis or color development in the fruit.

In leaves, *SlCLH2* and *SlCLH4* exhibited low and similar expression levels, and only a very small amount of *SlCLH2* expression was detected in mature fruits. Expression of *SlCLH3* was not detected in any of the samples analyzed. The expression of *SlCLH4* showed the highest in 5 DPA fruits, followed by a sharp decline in *SlCLH4* levels from 10 DPA to the mature green stage. The *SlCLH4* transcript was again highly induced by more than a 10-fold change from mature green to red mature. Based on the expression of the four *SlCLHs* in various tissues, *SlCLH1* was considered to be the predominantly expressed gene during leaf and fruit development/ripening stages (Figure 1B,C).

### 2.2. Various cis-Acting Elements Related to Abiotic Stresses and Phytohormone Response Are Present in the Promoters of Four SlCLH Genes

Using the *PlantCare* database [27], the distribution of *cis*-elements in the upstream 3 kb promoter region of four tomato *SlCLH* genes (*SlCLH1*–*4*) was analyzed, and the results are shown in Figure 1D. The results showed that in addition to the core *cis*-acting elements such as TATA-box and CAAT-box, 24, 22, 24, and 23 different *cis*-elements were arranged in the promoter regions of *SlCLH1*, *SlCLH2*, *SlCLH3*, and *SlCLH4* genes, respectively. These elements are considered to be involved in the responses to various different conditions, such as abiotic stresses (cold, drought, defense, and stress) and hormonal responses (ABA, MeJA, GA, IAA, SA, and ethylene). In this study, we found that there are four, three, one, and two *cis*-elements called ethylene-responsive elements in the promoters of *SlCLH1*-*4* genes, respectively.

Notably, approximately half of the *cis*-elements in the promoters of the four *SlCLH* genes were light-responsive elements (Figure 1D), suggesting that the *SlCLH* genes are modulated by light. To analyze the light response of *SlCLH* expression during fruit ripening, we designed a half-fruit experiment in which half of the green mature tomato fruit was exposed to continuous intense light (HL treatment) and the other half was shaded with silver paper (dark). In the HL treatment, *SlCLH1* transcript levels were significantly induced at 0.5 and 1 day compared to the equivalent maturation stage in dark treatment (Figure 1E).

### 2.3. Analysis of Amino Acid Sequences of SlCLHs

According to the genome sequence of tomato (the Tomato Genome Consortium, 2012), four *S. lycopersicum* CLH-encoding genes, referred to hereafter as *SlCLH1* (Solyc06g053980.2), *SlCLH2* (Solyc09g065620.2), *SlCLH3* (Solyc09g082600.1), and *SlCLH4* (Solyc12g005300.1) [21], were cloned and sequenced. The results revealed that the full-length cDNAs of *SlCLH1*, *SlCLH2*, *SlCLH3*, and *SlCLH4* were 939, 924, 954, and 1023 bp long, respectively. The deduced polypeptide sequences were 312, 307, 317, and 340 amino acids long with a calculated protein molecular mass of 34.05, 33.71, 34.60, and 37.26 kDa, respectively. A predicted pI of 6.86, 9.02, 8.17, and 9.14 of SlCLH1/2/3/4 is shown in Table 1. The estimated half-life of four SlCLHs is 30 h in vitro in mammalian reticulocytes, more than 20 h in vivo in yeast, and more than 10 h in vivo in *E. coli*. The instability index of SlCLH1 was computed to be 39.87, and the instability index of SlCLH2/3/4 was 44.59, 40.00, and 41.32, which classifies SlCLH1 as stable but SlCLH2/3/4 as unstable proteins. The calculated aliphatic index (93.08, 98.76, 90.98, and 86.03) indicates that SlCLHs are thermostable. The positive grand average hydropathy (GRAVY) values of SlCLH1/3/4 were found to range between −0.146 and −0.019, indicating the hydrophilic nature of these three proteins. However, the GRAVY value of SlCLH2 (0.081) indicates that it is a poorly hydrophilic protein. The N-terminal sequences of all four SlCLHs have no typical signal sequences for chloroplast localization predicted by SignalP-5.0 (Appendix A).

The deduced amino acid sequences of the four SlCLHs share 31.74–73.90% pairwise identity, while SlCLH1 shows the highest identity (73.90%) to SlCLH3. The SlCLHs have low homology with other plant CLHs (14.48–49.13%) used for pairwise analysis (Appendix A). The protein sequence of the four SlCLHs was aligned with other reported CLHs by using Clustal Omega. The result of multiple sequence alignment showed that all these CLHs share a highly conserved sequence GHSXG (Figure 2). In SlCLHs, the conserved GHSRG motif is also found in SlCLH1/3/4 which is in accordance with the result mentioned above; however, the conserved motif in SlCLH 2 is GHSKG.

The multiple sequence alignment also showed that the putative catalytic residues (serine (Ser)-aspartic acid (Asp)-histidine (His)) are strictly conserved in all selected chlorophyllases (marked with three yellow triangles in Figure 2). The putative catalytic triads of SlCLH1, SlCLH2, SlCLH3 and SlCLH4 may be Ser135-Asp162-His239, Ser132-Asp158-His234, Ser138-Asp168-His244, and Ser125-Asp172-His247, respectively.

The predicted tertiary structure of SlCLH1, SlCLH2, SlCLH3, and SlCLH4 were constructed with the SWISS-MODEL program using templates 5zoa.1.A of BTA-hydrolase 1 from *Thermobifida fusca*, 7ds7.1.A of Leaf-branch compost cutinase from *Biortu*, 6ilw.1.A of Poly (ethylene terephthalate) from *Ideonella sakaiensis*, and 5xh3.1.A of Poly (ethylene terephthalate) hydrolase from *Ideonella sakaiensis* 201-F6, respectively (Figure 3A–D). Based on the overall amino acid sequence, SlCLH1 shares 16.81% identity with the template 5zoa.1.A, SlCLH2 shares 16.81% identity with the template 7ds7.1.A, SlCLH3 shares 11.11% identity with the template 6ilw.1.A, and SlCLH4 shares 14.04% identity with the template 5xh3.1.A, respectively. The tertiary structural models are composed of ten β-strands and nine α-helices in SlCLH1, eight β-strands and nine α-helices in SlCLH2, eight β-strands and eleven α-helices in SlCLH3, eight β-strands and ten α-helices in SlCLH4, respectively (Figure 3F). The putative catalytic triads of SlCLH1 (Ser135-Asp162-His239), SlCLH2 (Ser132-Asp158-His234), SlCLH3 (Ser138-Asp168-His244), and SlCLH4 (Ser125-Asp172-His247) are located on a surface-exposed loop and clusters as the catalytic domain (Figure 3A–D). Active serine residues in SlCLHs are located in an extremely sharp turn between a β-strand and an α-helix (Figure 3E). Interestingly, the predicted tertiary structural model of four SlCLHs lacks the active site lid which is a common feature of the α/β hydrolase family.

### 2.4. Expression and Purification of Recombinant SlCLHs in E. coli

Previous studies have shown that the N-terminal peptide processing of CLHs plays a contradictory role in their Chl dephytylation activity. In order to develop further insights regarding the conservation of the CLH N-terminal processing events in the SlCLHs, the relevant CLH sequences (from *Citrus limon*, *Arabidopsis thaliana*, and *Solanum lycopersicum*) are aligned to find the potential processing sites (Figure 4A). Here, the N-terminal processing site of *S. lycopersicum* CLH (SlCLH1/2/3) was deduced by comparing the N-terminal protein sequence of other reported *CLHs*. The full-length CDS sequence of the four *SlCLHs* and N-terminally deleted *SlCLHs* versions (CLHs ΔN) was cloned into the pMal-p2x vector, to produce recombinant MBP-SlCLHs/CLHs ΔN fusion proteins in *E. coli*. After chemical induction of protein expression in *E. coli* Rosetta (DE3) cells, recombinant MBP-CLH1/2/3/4 proteins were present in the soluble protein fraction. The soluble recombinant SLCLH1/2/3/4 fused with an N-terminal MBP tag was purified with an amylose resin affinity column (NEB, E8021S, America). The SDS-PAGE result (Figure 4B) showed that the molecular weights of the full-length SlCLH1/2/3/4 and SlCLH1/2/3 ΔN corresponded to the calculated molecular masses of the MBP-tagged recombinant SlCLHs/SlCLHs ΔN. An in vitro SlCLHs activity assay was performed using purified full-length SlCLH1/2/3/4 and Chl as the substrate. We noticed that the MBP-SlCLH4 enzyme did not show any CLH activity (Figure 4C). Therefore, all subsequent experiments were performed with SlCLH1/2/3.

To evaluate the role of the predicted N-terminal sequences in the activity of recombinant SlCLHs, an in vitro CLH activity assay was performed using purified full-length and N-terminally truncated SlCLH1/2/3 and Chl a as the substrate. The results showed that full-length SlCLH1 and SlCLH2 showed significantly higher specific CLH activity than truncated SlCLH1 ΔN and SlCLH2 ΔN, while there was no difference in CLH activity in full-length SlCLH3 and SlCLH3 ΔN (Figure 4D). Removal of the predicted N-terminal processing sequence resulted in more than 80% and 40% loss of activity in SlCLH1 and SlCLH2, respectively, while the activity in SlCLH3 was still maintained. Therefore, the role of the predicted N-terminal processing sequences in the activity of recombinant SlCLHs remains puzzling.

### 2.5. Optimal Temperature and pH for the Activity of the Recombinant SlCLHs

The recombinant SlCLH1/2/3 had an optimal temperature for Chl hydrolysis of 30, 20, and 30 °C, and maintained > 70% of its peak activity in a temperature range of 20–40 °C for SlCLH1, 10–30 °C for SlCLH2, and 20–30 °C for SlCLH3. However, when the reaction temperature reached 60 °C, their CLH activity lost more than 70% (Figure 5A–C). After incubation for 24 h at 10 and 20 °C, more than 90% of SlCLH1/3 catalytic activity was retained. However, the catalytic activities of SlCLH1 and SlCLH2 dropped sharply after incubation at 30 °C and 20 °C for 24 h, and only 30% and 50% of the catalytic activities were retained. When the incubation temperature increased to 50 °C, only 4.1%, 14%, and 8.7% of SlCLH1/2/3 activity remained, respectively (Figure 5A–C). The above results suggest that the low catalytic activity of these enzymes at high temperature may be due to the changes in the conformation of these enzymes.

The optimum pH and pH stability of SlCLH1/2/3 were investigated in the pH range of 2–12. The optimum pH values of recombinant SlCLH1/2/3 were 9, 8, and 7, respectively, and about 80% of the peak activities of SlCLH1/2/3 were detected at pH 8–9, pH 8–10, and pH 7–9, respectively. The recombinant SlCLH1/2/3 were stable in the pH range of 8–9, 8–10, and 8–9, respectively. After incubation for 24 h at pH 6 and 11, the recombinant SlCLH1/2/3 residual activity was around 50–25%. While more than 95% of its catalytic activity was retained at pH 8, 9, or 10. (Figure 5D–F). In addition, the pH stability curves of recombinant SlCLH1/2/3 were similar to the optimum pH curves.

### 2.6. Enzyme Kinetics of Recombinant SlCLHs

To investigate the kinetic parameters and substrate specificity of recombinant SlCLHs, the initial reaction velocities (V_0_) of full-length recombinant SlCLHs toward different concentrations of three or two substrates (Chl a, Chl b, and Phein a) were determined (Appendix A). The kinetic parameters of recombinant SlCLHs, including maximal velocity (V_max_), Michaelis constant (K_m_), catalytic activity (K_cat_), and catalytic efficiency (K_cat_/K_m_), are summarized in Table 2. A comparison of the K_m_ and catalytic efficiency (K_cat_/K_m_) values of Chl a, Chl b, and Pheide a indicated that the recombinant SlCLH1 had the highest affinity and catalytic efficiency for Chl a (30.42 μM; 76.79 × 10^−5^ s^−1^μM^−1^), followed by Chl b (48.52 μM; 60.68 × 10^−5^ s^−1^μM^−1^), and the lowest affinity and catalytic efficiency for Phein a (302.50 μM; 1.06 × 10^−5^ s^−1^μM^−1^). Recombinant SlCLH1 preferentially hydrolyzes Chl a and Chl b over Phein a. Notably, Chl a, Chl b, and Phein a are all substrates of SlCLH1, but Phein a is not the substrate of SlCLH2 and SlCLH3. It suggests that different CLHs may be involved in different Chl breakdown pathways.

The catalytic efficiencies (K_cat_/K_m_) of SlCLH1/2/3 were also compared, and the results indicated that the recombinant SlCLH1 registered the highest catalytic efficiency of Chl a (76.79 × 10^−5^ s^−1^μM^−1^) and Chl b (60.68 × 10^−5^ s^−1^μM^−1^) with approximately 45- and 24-fold higher activity than those of SlCLH2 (Chl a, 1.72 × 10^−5^ s^−1^μM^−1^ and Chl b, 2.50 × 10^−5^ s^−1^μM^−1^) and almost 384 and 243 times the activity of SlCLH3 (Chl a, 0.20 × 10^−5^ s^−1^μM^−1^ and Chl b, 0.25 × 10^−5^ s^−1^μM^−1^), respectively.

### 2.7. Identification of Putative Catalytic Triad

In this study, the amino acid sequences of SlCLHs were aligned with four previously reported chlorophyllase sequences (from *Chlamydomonas reinhardtii*, *Citrus limon*, *Ginkgo biloba*, and *Arabidopsis thaliana*) (Figure 2). The putative catalytic residues in SlCLHs are marked with three yellow rectangles in Figure 2. The putative catalytic triads of SlCLH1, SlCLH2, SlCLH3, and SlCLH4 are Ser135-Asp162-His239, Ser132-Asp158-His234, Ser138-Asp168-His244, and Ser125-Asp172-His247, respectively.

Several site-directed mutagenesis studies have demonstrated the importance of the catalytic triad (Ser-Asp-His) for CLH activity. To test this, we generated three mutant versions of S135A, D162N, and H239A by site-directed mutagenesis to identify the role of these putative catalytic residues in SlCLH1 (Figure 6A,B), the major protein among the four SlCLHs. As shown in Figure 6C, the mutations of S135A, D162N, and H239A all resulted in a complete loss of activity compared to wild-type SlCLH1, indicating that Ser135, Asp162, and His239 residues are critical for the catalytic activity of SlCLH1.

## 3. Discussion

CLH plays a sole role in Chl degradation. The isolation and characterization of four *SlCLH* genes in tomato represent an important way toward understanding the biological function of SlCLHs during tomato leaf and fruit development/ripening stages.

### 3.1. Different Expression Pattern of SlCLHs Suggest That They May Have Distinct Roles in Various Tissues

In this study, *SlCLH* genes were cloned, and their expression profiles were analyzed by qRT-PCR. Compared to low expression of *SlCLH2/3/4* in leaves, the *SlCLH1* gene was mostly strong active in young and mature leaves while significantly decreased in aging leaves thereafter (Figure 1B), which was inconsistent with the results that CLH has no function in hydrolyzing Chl in senescing leaves [16]. *SlCLH1* exhibited the highest level of mRNA accumulation at 30 DPA and the mature green fruit stage and then declined during ripening (Figure 1C). The correlation between *SlCLH1* expression and Chl content confirmed the previous results that SlCLH1 was involved in Chl homeostasis or color development in fruit [21]. Regarding SlCLH2/3, the low levels of mRNA in fruit development and ripening described here is in accordance with the RNA-seq data available at Sol Genomics Network (http://solgenomics.net/; The data accessed on 26 May 2022). Notably, *SlCLH4* showed the highest expression level in 5 DPA fruits, followed by a sharp decline from 10 DPA to the mature green stage (Figure 1C). These differential expression patterns of *SlCLHs* suggest that they may engage in different biological functions.

In addition, Banas et al. reported that the expression level of *AtCLH1* and *AtCLH2* genes were increased upon illumination with white light [28]. Recent research confirmed that the transcript level of *AtCLH1* was significantly elevated after high light treatment to protect Arabidopsis young leaves from long-term light damage [11]. In this study, we found that approximately half of the *cis*-acting elements in the promoters of the four *SlCLH* genes were light-responsive elements (Figure 1D). The result of the half-fruit experiment showed that *SlCLH1* transcript level was significantly induced at 0.5 and 1 day in the high light treatment compared to the equivalent maturation stage in dark treatment (Figure 1E), which indicated that *SlCLH1* may have an additional role during fruit ripening.

### 3.2. SlCLHs Belong to the α/β Hydrolase Superfamily, but Show Different Physiological and Biochemical Characteristics

Previous studies have demonstrated the importance of the catalytic triad (Ser-Asp-His) for CaCLH activity [29]. In one of the studies, one catalytic residue (Ser-Asp-His) was replaced with other residues, and the activity of CLH was either reduced or completely lost [10,29,30]. In this study, introduction of three mutations into the putative catalytic triad of SlCLH1 (S135, D162, and H239) showed the complete loss of CLH activity (Figure 6C). Sequence alignment revealed that the amino acid residues in the four SlCLHs are conserved with the catalytic triad of other reported CLHs (Figure 2). This result strongly suggests that SlCLHs are similar to those of serine hydrolases.

The predicted tertiary structure of the four SlCLHs lacks the active site lid, which is a common feature of the α/β hydrolase family [31]. It has been shown that the absence of an active site lid in CLHs predicts the tertiary structure set for a rather wide substrate range. Chlorophyll, pheophytin, pheophorbide methyl ester, Zn pheophytin, 13^1^-hydroxy pheophorbide methyl ester, and Zn pyropheophytin are the substrate candidate of CLHs [2,30,32,33]. Kinetic analysis revealed that recombinant SlCLH1 has high hydrolysis activities against Chl a, Chl b, and pheophytin a (Phein a), but preferentially hydrolyzes Chl a and Chl b over Phein a, which is consistent with the preferences of both plant and algal CLHs [15,34]. Notably, Phein a is not the substrate of SlCLH2 and SlCLH3, while Chl a, Chl b, and Phein are all not the substrate of SlCLH4. SlCLH4 may have some other substrate specificity or enzyme function other than SlCLH1/2/3. Further molecular and biochemical investigations are needed to elucidate the actual function of SlCLH4. Taken together, different characteristics between SlCLHs implied that they might have different physiological roles in Chl degradation in tomato.

### 3.3. The Role of the Predicted N-Terminal Processing Sequences in the Activity of Recombinant SlCLHs

Previous studies have shown that the N-terminal peptide processing of CLHs plays a contradictory role in their Chl dephytylation activity. For example, the chlorophyllases from *Pachira macrocarpa* and *Cyanobacterium Oscillatoria cuminata* PCC6304 were functionally inactive when the N-terminal peptide was removed [8,10]. However, CLH lacking the N-terminal 21 amino acids from *Citrus sinensis* showed more enzymatic activity in vivo [35]. In order to develop further insights regarding the conservation of the CLH N-terminal processing events in the SlCLHs, the relevant CLH sequences (from *Citrus limon*, *Arabidopsis thaliana*, and *Solanum lycopersicum*) are aligned to find the potential processing sites (Figure 4A). Here, the N-terminal processing site of *S. lycopersicum* CLH (SlCLH1/2/3) was deduced by comparing the N-terminal protein sequence of *C. limon* CLH [6,7]. In this research, removal of the predicted N-terminal processing sequence resulted in more than 80% and 40% loss of activity in SlCLH1 and SlCLH2, respectively, while the activity in SlCLH3 was still maintained (Figure 4D). Therefore, the predicted N-terminal processing sequences of recombinant SlCLH1/2 is essential for their activities, but it is not for SlCLH3. These contradictory results in SlCLH isozymes led to a debate about the possible role of the predicted N-terminal processing sequences in the activity of recombinant SlCLHs. Further validation is required in future studies.

## 4. Materials and Methods

### 4.1. Plant Materials

Tomato (*Solanum lycopersicum* cv Micro-Tom/Ailsa Craig) were grown in pots (140 × 130 mm; 330 × 290 mm, respectively) with sterilized soil (Jiffy7^®^, Jiffy Products International B.V.) at the chamber under the following conditions: 16/8 h light/dark at 25/18 ± 2 °C (250 µmol m^−2^s^−1^) and 60–70% relative humidity. The tomato plants were fed with fertilizer (N-P_2_O_5_-K_2_O:15-6-8) every two weeks. For development studies, the first true leaf at 20 d after germination and the leaves from the middle plant location at 50 ± 2 and 80 ± 2 d after germination were selected as young, mature, and senescence stages, respectively, and fruits tagged at anthesis (DPA, days post anthesis) were collected at 5, 10, 20, and 30 DPA. For ripening studies, tomato fruits at mature green (MG), breaker (BR), pink, and red ripe stage were collected.

Half-fruit experiment: half of the mature green tomato fruit (*S. lycopersicum* cv Ailsa Craig) was exposed to continuous intense light (680 µmol m^−2^s^−1^, HL treatment) and the left half fruit was shaded with silver paper (dark treatment). The tomato pericarp from the HL and dark treatment was collected at 0.5, 1, 2, 3, and 4 days, respectively. Samples were immediately frozen in liquid nitrogen and stored at −80 °C until use. A minimum of three biological replicates were used for each experiment.

### 4.2. RNA Isolation

Total RNA was isolated from 1 mg of tomato leaves or fruit pericarps with a TRIzol reagent (Invitrogen, Carlsbad, California, America), and cDNA was synthesized with a PrimeScript cDNA Synthesis Kit (TaKaRa, Shiga, Japan) following the manufacturer’s manuals.

### 4.3. Gene Expression Analysis

The expression levels of genes were quantified by qRT-PCR using gene-specific primers (Appendix A) with ChamQ Universal SYBR qPCR Master Mix (Vazyme Biotech Co., Ltd. Nanjing, China) on a CFX Connect^TM^ Real-Time System (Bio-Rad, Hercules, California, America). A PCR reaction mix contained 10 μL of ChamQ Universal SYBR qPCR Master Mix (Vazyme Biotech Co., Ltd. Nanjing, China), 125 nM of each forward and reverse primer, 2.0 μL of cDNA template, and 7.5 μL of RNase-free water. Data analysis was calculated by the 2^−ΔCt^ method or 2^−ΔCtΔCt^ method [36]. All experiments were measured in three biological replicates.

### 4.4. Protein Expression and Purification

According to the genome sequence of tomato (*Solanum lycopersicum*), PCR primers listed in Appendix A were designed and used to amplify the full-length CDS region of *SlCLH1* (Solyc06g053980.2), *SlCLH2* (Solyc09g065620.2), *SlCLH3* (Solyc09g082600.1), and *SlCLH4* (Solyc12g005300.1) genes. The CDS of *CLHs* were amplified with KOD DNA polymerase (Toyobo Life Science, Osaka, Japan) by PCR. After that, the endonuclease sites *Bam*H I and *Sal* I were introduced at the 5′- and 3′-termini of the target gene, respectively, to allow the insertion of the target gene into the *Bam*H I/*Sal* I predigested pMal-p2x vector with an N-terminal fusion of maltose-binding protein (MBP). N-terminally truncated Chlases versions were also constructed as follows: 939 bp CDS of *SlCLH1*, lacking the first 36 codons; 924 bp CDS of *SlCLH2*, lacking the first 39 codons; 954 bp CDS of *SlCLH3*, lacking the first 54 codons. All these truncated CLH ΔN versions were supplemented with an ATG start codon. The primer pairs for amplifying and generating MBP-CLHs and MBP-CLHs ΔN were listed in Appendix A. After verification of sequence accuracy by sequencing, MBP-CLHs/CLHs ΔN were introduced into *E. coli* Rosetta (DE3) cells for protein expression and subsequent purification.

The recombinant *E. coli* Rosetta (DE3) cells were incubated at 16 °C for about 24 h with 0.5 mM isopropyl *β*-D-1-thiogalactopyranoside (IPTG). Bacteria were pelleted by centrifugation and resuspended in 25 mL lysis buffer (50 mM Tris-HCl, 100 mM NaCl, 5% Glycerin, 1% (*w*/*v*) Protease Inhibitor Cocktail, pH 7.4). Cells were continuously lysed by sonication for 30 min. After that, samples were centrifuged at 10,000× *g* for 30 min at 4 °C to separate the soluble and pellet fractions. The MBP-CLHs/CLHs ΔN soluble fractions were purified using an amylose resin affinity column (NEB, E8021S, Ipswich, Massachusetts, America). The protein concentration of recombinant SlCLHs was quantified using the Bradford assay method (Modified Bradford Protein Assay Kit, SANGON Biotech, Shanghai China). The presence of expressed proteins was analyzed by SDS-PAGE.

### 4.5. Identification of the Putative Catalytic Triad

The mutant enzymes Ser135Ala, Asp162Asn, and His239Ala of SlCLH1 were constructed to identify the role of these putative catalytic residues in the activity of SlCLH1. The desired mutations were introduced into the Mal-p2x vector with site-directed mutagenesis using the PCR overlap extension method [10]. The primers listed in Appendix A were designed to introduce the desired mutations. DNA sequencing was performed to confirm the accuracy of the gene encoding the mutated enzyme. Cultivation of these transformants and expression and purification of the mutant enzymes were performed as described above for the wild-type enzymes.

### 4.6. Sequence Analysis

The putative protein sequence of *Solanum lycopersicum* chlorophyllases (SlCLHs), SlCLH1 (Solyc06g053980.2), SlCLH2 (Solyc09g065620.2), SlCLH3 (Solyc09g082600.1), and SlCLH4 (Solyc12g005300.1) were obtained from the International Tomato Genome Sequencing Consortium (SGN; solgenomics.net) database (version ITAG 2.4). The molecular weight, theoretical pI, estimated half-life, instability index, aliphatic index, and GRAVY value were predicted with the ProtParam tool (https://web.expasy.org/protparam/ The data accessed on 10 April 2022).

A multiple sequence alignment of the putative four *S. lycopersicum* chlorophyllases and other published chlorophyllases was constructed with Clustal Omega (https://www.ebi.ac.uk/Tools/msa/clustalo/; The data accessed on 29 January 2022). A homology structural model of SlCLHs was constructed with the SWISS-MODEL server (https://swissmodel.expasy.org; The data accessed on 4 January 2022). The iCn3D server was used to verify and validate the three-dimensional structure of SlCLHs (https://www.ncbi.nlm.nih.gov/Structure/icn3d/full.html; The data accessed on 4 January 2022). The putative SlCLHs sequence was also analyzed with the program SignalP 5.0 server (http://www.cbs.dtu.dk/services/SignalP/; The data accessed on 2 June 2022).

### 4.7. In Silico Analysis of Cis-Elements in Promoters of SlCLH Genes

The promoter sequences (≈3 kb of the 5′ upstream region of the start codon) of the four *SlCLH* genes were extracted from the International Tomato Genome Sequencing Consortium (SGN; http://www.solgenomics.net) database (version ITAG 2.4). The data accessed on 7 August 2022. The PlantCARE relational database [27] was used for plant cis-element searches in the promoter of these *SlCLH* genes (Figure 1D).

### 4.8. Chl Dephytylation Activity Assay

Chl dephytylation activity was measured according to the method described by Arkus et al. [37]. Chlorophylls isolated from spinach leaves were used for enzyme assays; however, an enzyme kinetics assay for the comparison of the activity of SlCLH1/2/3/4 was performed by using Chl a and Chl b as standard substrates purchased from Sigma-Aldrich (St. Louis, MO, USA). Cells containing the empty vector pMal-p2x were used as a control. The 100 μL reaction buffer (100 mM sodium phosphate and 0.24% Triton X-100 (pH8.0)) contained 0.5 μg of SlCLH1 or 5 μg of SlCLH2/3/4, and 100 μM of substrates dissolved by acetone. The reaction was incubated at 30 °C for 1 h (SlCLH1 and SlCLH2) or 4 h (SlCLH3), then stopped by adding 110 μL of a stop buffer (4:6:1 (*v*/*v*) acetone/*n*-hexane/10 mM KOH). The reaction mixture was vigorously vortexed and centrifuged at 12,000× *g* for 5 min to obtain two separate layers (or phases): the upper layer contained the unreacted Chl in *n*-hexane, and the lower layer contained the chlorophyllide (Chlide) product in an aqueous acetone solution.

The absorbance of the aqueous acetone layer was measured at 667 nm for Chlide a, 651 nm for Chlide b, and 667 nm for pheophorbide a (Pheide a) by using a spectrophotometer. Millimolar extinction coefficients of 76.79, 47.04, and 47.2 mM^−1^cm^−1^ were employed for calculating the generation amount of Chlide a and Chlide b, respectively. One unit of enzyme activity was defined as the amount of enzyme required to catalyze the generation of 1 μmol Chlide a, Chlide b, or Pheide a per minute at 30 °C. The specific activity of the enzyme was defined as the enzyme activity (units) per milligram of protein. All enzymatic assays were performed in triplicate.

### 4.9. Analysis of Biochemical Characteristics of Recombinant SlCLHs

The effects of pH and temperature on the enzyme activity of recombinant SlCLHs were determined by measuring its Chl dephytylation activity as mentioned above. The optimal pH was investigated by preparing reaction buffers in the pH range of 2.0–12.0 containing 100 mM sodium phosphate buffer and 0.24% Triton X-100. The pH was adjusted by adding HCl or NaOH. To analyze the pH stability of the recombinant SlCLHs, the enzyme solutions were incubated in reaction buffers (containing 0.24% Triton X-100 (pH 2.0–12.0)) at 4 °C for 24 h. Enzyme activity was subsequently measured using the aforementioned method.

To determine the effect of temperature on the enzyme activity of recombinant SlCLHs, reactions were performed at temperatures ranging from 10 to 90 °C for 24 h. To analyze thermal stability, the enzyme solutions were preincubated at temperatures ranging from 10 to 90 °C for 24 h and subsequently cooled before activity analysis. All reactions were performed in triplicate. The resulting average values and standard deviations are presented. The highest activity is presented as 100%, with other activities presented as percentages relative to the highest activity.

### 4.10. Enzyme Kinetic Assay

Kinetic assays of purified recombinant SlCLHs were detected with a spectrophotometer. SlCLH1 and substrate Chl a and Chl b were performed for 0.5 h at 30 °C and 4 h for Phein a. SlCLH2 and substrate Chl a and Chl b were performed for 2 h at 25 °C, SlCLH3 and substrate Chl a and Chl b were performed for 4 h at 30 °C. The initial reaction velocity (V_0_) was determined with different concentrations of substrates in the reaction buffer (100 mM sodium phosphate and 0.24% Triton X-100, pH 7.0). Eight concentrations (10, 20, 40, 60, 80, 100, 120, and 140 μM) of Chl a and Chl b and nine concentrations (40, 60, 80, 100, 120, 200, 300, 400, and 500 μM) of Phein a were used for enzyme kinetics of SlCLH1. Nine concentrations (40, 60, 80,100, 120, 150, 300, 400, and 500 μM) of Chl a and Chl b were used for enzyme kinetics of SlCLH2. Nine concentrations (40, 60, 80, 100, 120, 300, 400, 500, and 600 μM) of Chl a, nine concentrations (40, 60, 80,100, 120, 150, 200, 400, and 500 μM) of Chl b were used for enzyme kinetics of SlCLH3. The production of Chlide a, Chlide b, and Pheide a was measured as the maximum absorbance of each product (measured at 667, 651, and 667 nm, respectively). We calculated the kinetic parameters K_cat_ and K_m_ according to the Michaelis–Menten kinetics for nonlinear regression (Origin software, version 6.1, Northampton, MA, USA) and generated a plot of reaction velocity versus substrate concentration.

## 5. Conclusions

Four SlCLHs contain a conserved GHSXG lipase motif and a Ser-Asp-His catalytic triad that belong to the α/β hydrolase superfamily. Based on the expression of these four genes in various tissues, *SlCLH1* was considered to be the predominant *CLH* gene during leaf and fruit development/ripening stages. The results of promoter analysis and gene expression indicate that the activity of *SlCLH1* can be modulated by light. The N-terminal peptide is essential for the activity of SlCLH1/2. In this study, recombinant SlCLH1 could hydrolyze Chl a, Chl b, and Phein a, but preferred Chl a and Chl b over Phein a, whether it can hydrolyze other similar substrates needs to be investigated later. The recombinant SlCLH1 is more efficient for Chl dephytylation than SlCLH2, SlCLH3, and SlCLH4. These different characteristics between SlCLHs suggest that they may have different physiological functions during Chl breakdown in tomato.

## Figures and Tables

**Figure 1 ijms-23-11716-f001:**
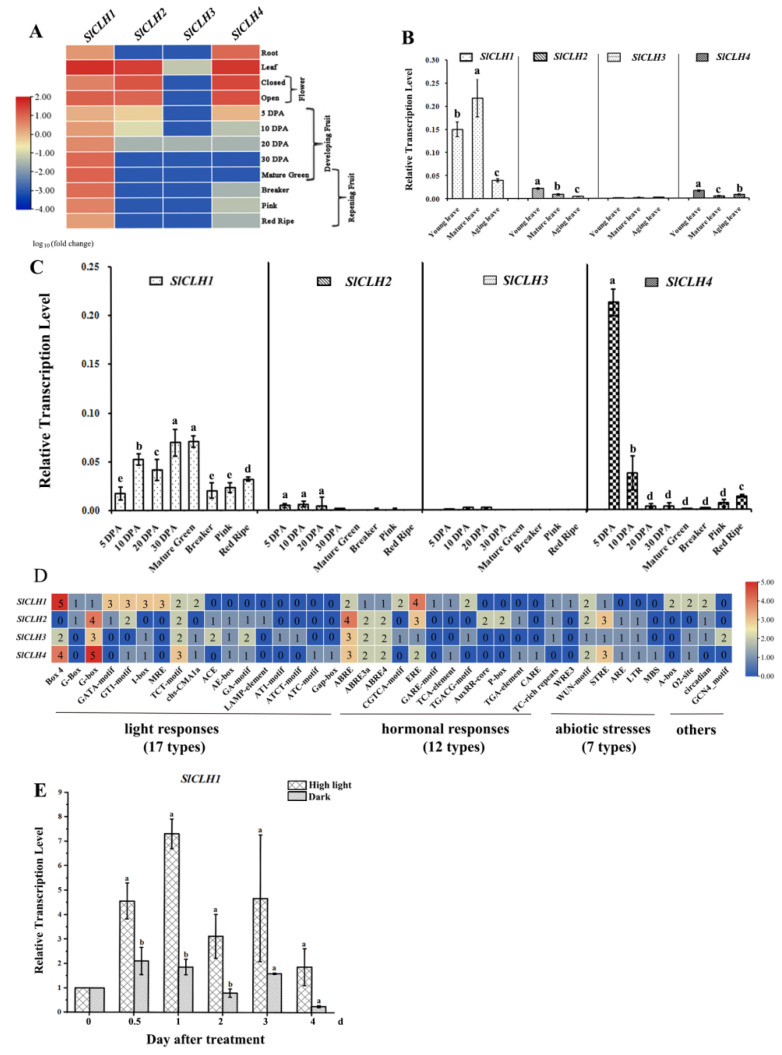
Expression profiles of four *SlCLHs*. (**A**) Relative expression values for *SlCLHs* in plant development and fruit ripening (root, leaf, bud, flower, 4 fruit developmental stages, mature green (MG), breaker (BR), pink and red ripe (BR + 10)) was calculated Log_10_ RPKM values derived from RNA-seq data from the SGN database (http://ted.bti.cornell.edu/cgi-bin/TFGD/digital/search.cgi?ID=D004, *S. lycopersicum* cv. Heinz; https://tea.solgenomics.net/expression_viewer/input, *S. lycopersicum* M82); The date accessed on 26 May 2022. (**B**,**C**) Quantitative real-time PCR of *SlCLH* genes in different leaf development (**B**) and fruit development/ripening stages (**C**). The relative transcript levels were displayed as expression ratio between *SlCLH* genes and the reference gene. (**D**) *Cis*-acting element analysis of the promoter region of *SlCLHs*. The *cis*-elements located in the 3000 bp promoter sequence (upstream of the start codon) of all *SlCLHs* were analyzed via the *PlantCARE* website. The number of corresponding *cis*-acting elements was used for the heatmap construction. (**E**) Changes in *SlCLH1* transcript levels in mature green tomato, which responded to continuous high light and dark treatment, respectively. The samples were collected at 0.5, 1, 2, 3, and 4 days after high light treatment. Data represent means ± the standard deviation of three biological repeats. One-way analysis of variance (ANOVA) with Tukey’s method was applied to show the significant differences in (**B**,**C**,**E**) at *p* < 0.01 in pair-wise comparison and classified with the letter a–e.

**Figure 2 ijms-23-11716-f002:**
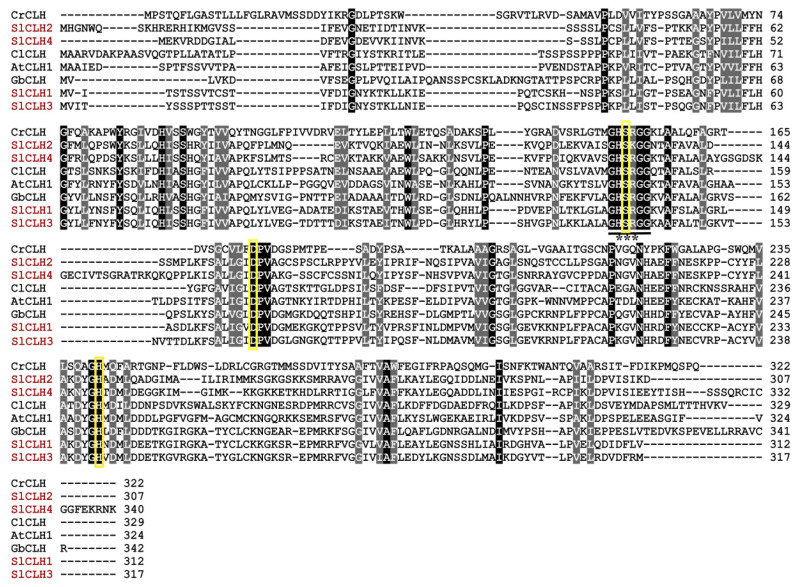
Alignment of amino acid sequences of various CLHs. The sequences of CLHs from *Arabidopsis thaliana* (*AtCLH1*), *Ginkgo biloba* (*GbCLH*), *Citrus limon* (*ClCLH1*), and *Chlamydomonas reinhardtii* (*CrCLH*) were aligned using Clustal Omega. Asterisks represent lipase motif. The identical amino acids in all CLHs are shaded in black, and identical amino acids in seven out of eight sequences are shaded in gray. The black line represents the conserved GHSXG lipase motif, and the three yellow boxesdenote the putative catalytic triad: Ser, Asp, and His residues.

**Figure 3 ijms-23-11716-f003:**
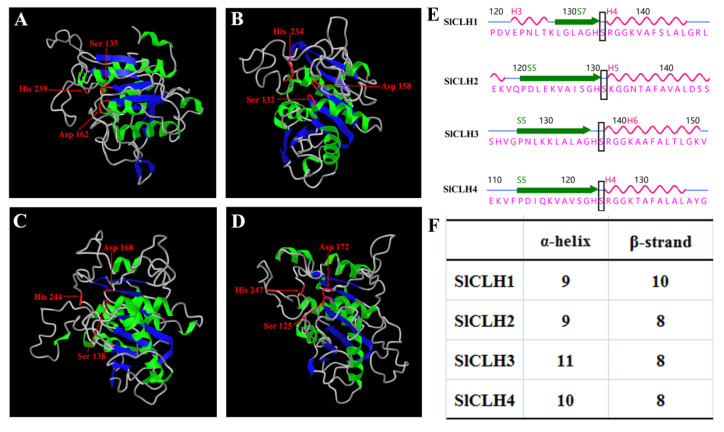
Predicted tertiary structure of four SlCLHs. This diagram was constructed with the SWISS-MODEL server. The conserved catalytic triad residues of SlCLH1 (**A**), SlCLH2 (**B**), SlCLH3 (**C**), and SlCLH4 (**D**) are indicated. (**E**) The number of stands and helices in the tertiary structure. The location of active serine residue in SlCLHs, which is usually at the extremely sharp turn between a β-strand and an α-helix, called the ‘nucleo-philic elbow’, a conserved structure in the α/β hydrolases. (**F**) The number of β-strands and α-helices in SlCLHs.

**Figure 4 ijms-23-11716-f004:**
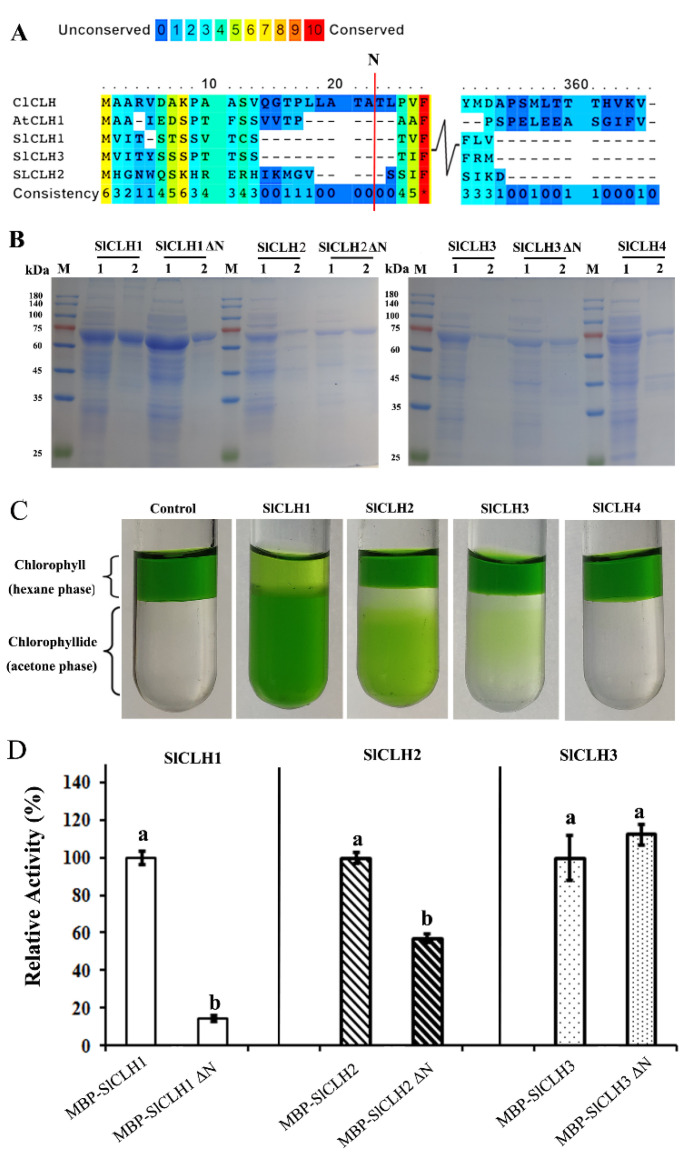
Effect of the predicted N-terminal processing sites on the activity of recombinant SlCLHs. (**A**) Alignment of partial N- and C-terminal amino acid sequences of SlCLHs with *Arabidopsis thaliana* (*AtCLH1*) and *Citrus limon* (*ClCLH1*). The putative N-terminal processing sites of SlCLHs are indicated by a red line. (**B**) SDS-PAGE analysis of N-terminally MBP-tagged recombinant SlCLHs and SlCLHs ΔN. M, prestained molecular weight marker. Line 1, crude extract of recombinant SlCLHs. Lane 2, purified recombinant SlCLHs after amylose resin affinity column chromatography. The protein gel was stained with Coomassie Brilliant Blue. (**C**) Chlorophyll and Chlide were separated by an acetone/hexane phase separation assay. After incubation for 60 min, Chl substrate remained in the hexane phase in the empty vector solution, while the Chlide product partitioned into the acetone phase in activity CLH solution. (**D**) Relative activity in recombinant SlCLHs and SlCLH ΔN. Specific activities of purified full-length and truncated SlCLHs were measured with the standard CLH activity assay method, using Chl a as the substrate. The Bradford method was used to quantify the protein. Values are the means ± SD of three independent experiments. One-way analysis of variance (ANOVA) with Tukey’s method was applied to show the significant differences in (**B**,**C**,**D**) at *p* < 0.01 in pair-wise comparison and classified with the letter a or b.

**Figure 5 ijms-23-11716-f005:**
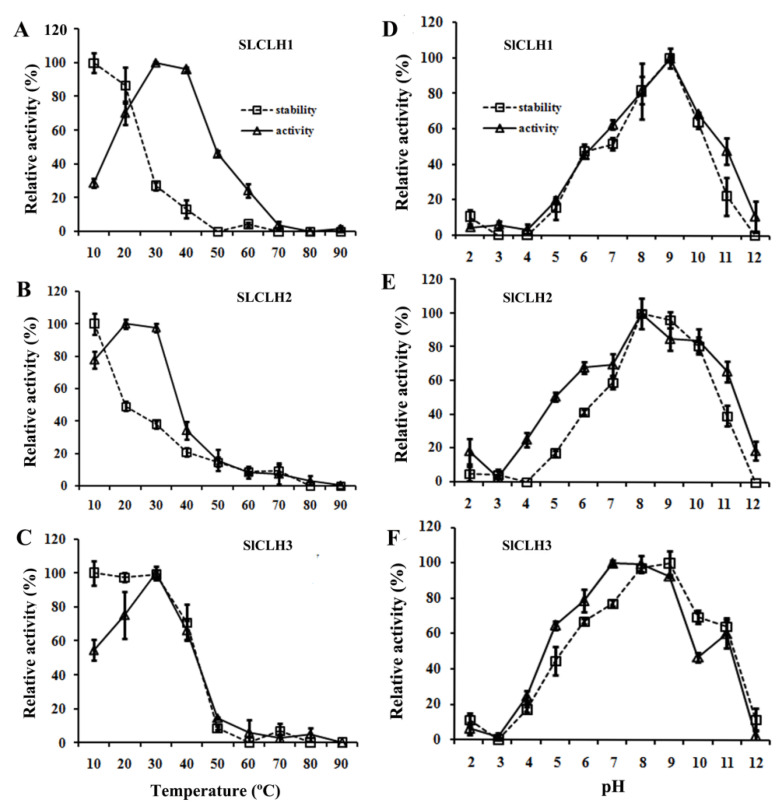
Effect of temperature and pH on the activity and stability of recombinant SlCLH1/2/3 Effect of temperature on the activity and stability of recombinant SlCLH1 (**A**), SlCLH2 (**B**), and SlCLH3 (**C**). The optimal temperature (△) and temperature stability (□) of SlCLH1, SlCLH2, and SlCLH3 were measured according to the standard CLH assay by using Chl a as a substrate. Effect of pH on the activity and stability of recombinant SlCLH1 (**D**), SlCLH2 (**E**), and SlCLH3 (**F**). The optimal pH (△) and pH stability (□) of SlCLH1, SlCLH2, and SlCLH3 were measured according to the standard CLH assay by using Chl a as a substrate. Data represent means ± the standard deviation of three independent experiments.

**Figure 6 ijms-23-11716-f006:**
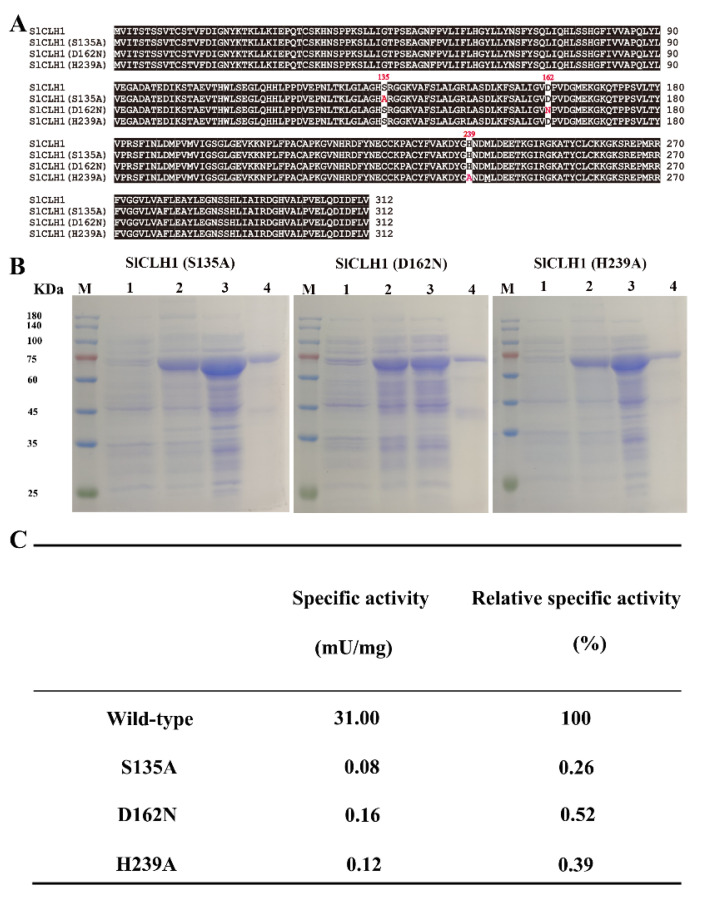
Confirmation of the catalytic triad of SlCLH1 by Site-directed mutation. (**A**) Alignment of amino acid sequences of wild-type SlCLH1 and the generated SlCLH1 sequences with site-directed mutation at the catalytic triad. Three mutated SlCLH1 were generated with one amino acid mutation of the catalytic triad residues for each. Site-directed mutagenesis introduced in three mutated SlCLHs are shown in red. (**B**) SDS-PAGE analysis of N-terminally MBP-tagged recombinant mutant SlCLH1. M, prestained molecular weight marker. Line 1, crude extract of recombinant SlCLHs. Lane 2, purified recombinant SlCLHs after amylose resin affinity column chromatography. The protein gel was stained with Coomassie Brilliant Blue. (**C**) Specific activities and relative activities of purified wild-type and mutated SlCLH1. The relative specific activity of wild-type SlCLH1 was deemed to be 100%. The specific activity of the enzyme was defined as the enzyme activity (units) per milligram of protein.

**Table 1 ijms-23-11716-t001:** Physical and biochemical parameters of SlCLHs predicted with the ProtParam tool.

Source	Accession No.	Molecular Weight	pI	Estimated Half-Life (Hours)			Instability Index	Aliphatic Index	GRAVY
				Mammalian Reticulocytes, In Vitro	Yeast, In Vivo	*E. coli*, In Vivo			
SlCLH1	Solyc06g053980.2.1	34,048.30	6.86	30 h	>20 h	>10 h	39.87	93.08	−0.019
SlCLH2	Solyc09g065620.2.1	33,706.42	9.02	30 h	>20 h	>10 h	44.59	98.76	0.081
SlCLH3	Solyc09g082600.1.1	34,600.81	8.17	30 h	>20 h	>10 h	40.00	90.98	−0.028
SlCLH4	Solyc12g005300.1.1	37,257.12	9.14	30 h	>20 h	>10 h	41.32	86.03	−0.146

**Table 2 ijms-23-11716-t002:** Substrate specificity and kinetic parameters of recombinant SlCLHs.

Recombinant SlCLHs	Substrate	Vmax	K_m_ (μM)	k_cat_ (10^−3^ s^−^^1^)	k_cat_/K_m_
		(10^−3^ μmoL mg^−1^ min^−1^)			(10^−5^ s^−^^1^μM^−^^1^)
SlCLH1	Chl a	18.93	30.42	23.36	76.79
	Chl b	23.86	48.52	29.45	60.68
	Phein a	2.61	302.50	3.22	1.06
SlCLH2	Chl a	1.96	140.17	2.41	1.72
	Chl b	2.22	109.03	2.73	2.50
SlCLH3	Chl a	0.41	258.86	0.51	0.20
	Chl b	0.59	295.32	0.73	0.25

## Data Availability

“MDPI Research Data Policies” at https://www.mdpi.com/ethics (accessed on 25 September 2022).

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
