# Peer review of "Genes, Structural, and Biochemical Characterization of Four Chlorophyllases from Solanum lycopersicum"

_ijms, 2022, doi:10.3390/ijms231911716_

Round 1

Reviewer 1 Report

Main comments:

The authors of this scientific study have addressed a little-known topic related to photosynthetic pigments. The manuscript is well, carefully prepared, with clear figures and tables that are very well described. After completing the description in Materials and Methods, the manuscript, in my opinion, is most deserving of publication.

Detailed comments and suggestion:

Materials and Methods

Line 404: complete the information on the growth of the tomato in the chamber, complete what the tomatoes were grown in (in pots or maybe containers?) in soil, peat or other substrate, whether the tomato was fed with nutrient or fertilizer.

Line 406: which tomato leaves were taken for testing lower, middle or maybe upper? Preferably also days after sowing.

Author Response

Dear reviewer,

Thank you very much for reviewing our manuscript ijms-1936521. Your professional comments and suggestions made great help to improve the quality of this manuscript. According to your comments, we have done some revisions.

Materials and Methods

Line 404: complete the information on the growth of the tomato in the chamber, complete what the tomatoes were grown in (in pots or maybe containers?) in soil, peat or other substrate, whether the tomato was fed with nutrient or fertilizer.

Response: Thank you for the suggestion. We have added relevant information in the resubmitted manuscript. The tomato in this study were grown in pots (140*130 mm; 330*290 mm, respectively) with sterilized soil (Jiffy7®, Jiffy Products International B.V.) at the chamber under the following conditions: 16/8 h light/dark at 25/18±2 °C (250 µmol m-2 s-1) and 60-70 % relative humidity. The tomato plants were fed with fertilizer (N-P2O5-K2O: 15-6-8) every two weeks.

Line 406: which tomato leaves were taken for testing lower, middle or maybe upper? Preferably also days after sowing.

Response: Thank you for the question. We have added relevant information in the resubmitted manuscript. For development studies, the first true leaf at 20 d after germination and the leaves from the middle location of plant at 50±2 and 80±2 d after germination were selected as young, mature, and senescence stages. We have added the detail information in 4.1 Plant Material.

Reviewer 2 Report

The manuscript Genes, structural and biochemical characterization of four chlorophyllases from Solanum lycopersicum by Guangyuan Liu, Xue Meng, Yujun Ren, Min Zhang, Ziqing Chen, Zhaoqi Zhang and Xuequn Pang and Xuelian Zhang discusses an important aspect of how enzymes work.

Extremely unfortunate expression: Recombinant SlCLH1 could hydrolyze Chl a, Chl b, and Phein a, but preferred to use Chl a and Chl b as substrates. Since the conscious choice of substrates is an extremely embarrassing idea.

The conclusion should also be rewritten so that the important results obtained in the manuscript are presented in a more practical way.

Author Response

Dear reviewer,

Thank you very much for reviewing our manuscript ijms-1936521. Your professional comments and suggestions made great help to improve the quality of this manuscript. According to your comments, we have done some revisions.

Extremely unfortunate expression: Recombinant SlCLH1 could hydrolyze Chl a, Chl b, and Phein a, but preferred to use Chl a and Chl b as substrates. Since the conscious choice of substrates is an extremely embarrassing idea. The conclusion should also be rewritten so that the important results obtained in the manuscript are presented in a more practical way

Response:Thank you for the suggestion. We have changed relevant information in the resubmitted manuscript. We revised the sentence "Recombinant SlCLH1 could hydrolyze Chl a, Chl b, and Phein a, but preferred to use Chl a and Chl b as substrates" to "In this study, recombinant SlCLH1 could hydrolyze Chl a, Chl b, and Phein a, but preferred to Chl a and Chl b, whether it can hydrolyze other similar substrates needs to be investigated later"